# Evolutionary Insights of Hepatitis B Virus Genotypes and Profiles of Mutations in Surface and Basal Core Promoter/Pre-Core Genes Among HBsAg-Positive Patients in North-Central and Southwestern Nigeria

**DOI:** 10.3390/v17081101

**Published:** 2025-08-10

**Authors:** Priscilla Abechi, Uwem E. George, Olawale A. Adejumobi, Umar Ahmad, Olamide Y. Aborisade, Arthur O. Oragwa, Oluremi I. Ajayi, Oluwasemilogo O. Akinlo, Christian Happi, Onikepe A. Folarin

**Affiliations:** 1Institute of Genomics and Global Health (Formerly ACEGID), Redeemer’s University, Ede 232101, Nigeria; george27@run.edu.ng (U.E.G.); adejumobi13779@run.edu.ng (O.A.A.); ajayi9887@run.edu.ng (O.I.A.); akinlooluwasemilogo@gmail.com (O.O.A.); happic@run.edu.ng (C.H.); 2Department of Biological Sciences, Faculty of Natural Sciences, Redeemer’s University, Ede 232101, Nigeria; 3Institute of Genomics, Centre for Laboratory Diagnostics and Systems, Africa Centres for Disease Control and Prevention (Africa CDC), African Union Comission, Ababa 3243, Ethiopia; umara@africacdc.org; 4Haematology and Blood Transfusion Service Department, UNIOSUN Teaching Hospital, Osogbo 230283, Nigeria; yettymama2011@gmail.com; 5Department of Veterinary Microbiology, Faculty of Veterinary Medicine, University of Jos, Jos 930003, Nigeria; oragwaa@unijos.edu.ng

**Keywords:** immune escape mutants, precore/basal core promoter region, Northcentral/Southwestern Nigeria, HBV genotype

## Abstract

In Nigeria, hepatitis B virus (HBV) infection remains a significant public health issue. The emergence of immune escape mutants (IEMs), basal core promoters, and precore (BCP/PC) mutants among asymptomatic individuals has enabled the continuous evolution of the virus in the country. In this study, we used Sanger sequencing of the S gene and the BCP/PC region to investigate the genetic diversity, phylogenetic relationships, and mutational profiles of HBV strains detected in two regions in Nigeria. A total of 178 HBsAg-positive samples confirmed by ELISA underwent viral DNA extraction and PCR amplification of the surface and BCP/PC genes, and 76 and 60 sequences were found to be exploitable for S and BCP/PC genes, respectively, which were used for HBV genotyping and mutational analysis. We detected various mutations in the major hydrophilic loop (target of neutralizing antibodies), including vaccine escape mutants (VEMs) (L127P/R, S140T/L, and G145A), HBV immunoglobulin resistance mutants (T131N, S143T, and W156R), and mutations previously reported in patients with reactivated infections (T115N, G159A/R, and F161Y). We also identified a high proportion of C1741T in 34/42 (81%) along with A1762T or G1764A mutation in 14/42 (33%) and 18/42 (43%) as the dominant variants in the BCP region. The predominant classical PC G1896A and G1899A variants were identified in 26/42 (62%) and 17/42 (40%) participants in this study. Two HBV genotypes were identified (A and E). However, HBV genotype E was the most frequently identified genotype, and is still the dominant strain circulating in Nigeria. We report the circulation of HBV IEMs and the preponderance of BCP and classical PC variants among asymptomatic carriers. Our findings suggest that the spread of these HBV mutant variants among asymptomatic carriers may have an impact on the effectiveness of diagnostic immunoassays and the success of HBsAg-based vaccinations. This highlights the need for robust surveillance.

## 1. Introduction

Hepatitis B virus (HBV) continues to be a global public health challenge, with over 296 million individuals chronically infected worldwide [1]. The virus is one of the leading causes of liver diseases, including cirrhosis and hepatocellular carcinoma (HCC), contributing significantly to morbidity and mortality [2,3]. Despite the availability of effective vaccines, HBV remains endemic in many regions, with sub-Saharan Africa experiencing some of the highest prevalence rates globally [4,5,6]. Horizontal and perinatal transmission contributes to a high prevalence of chronic infections in these regions [7,8,9].

The HBV genome is made up of approximately 3.2 kb of relaxed-circular DNA (rcDNA), which encodes four overlapping open reading frames (ORFs): core (C), polymerase (P), surface (S), and X. The four ORFs encode seven proteins: pol, X (HBx), precore, core, and three envelope proteins (surface antigens): L-HBs, M-HBs, and S-HBs [10]. Genetic diversity is a distinguishing feature of HBV, with the virus divided into ten genotypes (A–J) based on the sequence divergence of at least 8% across the whole genome [11]. These genotypes exhibit different responses to antiviral treatment, clinical outcomes, transmission pathways, and geographic distributions. For example, genotype E has been linked to HBV drug resistance and immune evasion [12,13], whereas genotype A is known to be common in HBe-negative chronic active hepatitis HBV [14]. Individuals infected with genotypes C and D have a higher risk of severe liver disease (HCC) than those infected with genotypes A and B [15]. Genotype E is generally found in sub-Saharan Africa, particularly in West Africa [16,17,18,19], and has low genetic diversity when compared to other HBV genotypes, which is critical for addressing challenges such as vaccine escape mutations, diagnostic reliability, and therapeutic strategy development [11]. Maternal–fetal transmission of HBV genotype E is estimated to account for 8% of chronic HBsAg carriers [16]. Despite its regional dominance, Genotype E remains poorly understood, particularly in terms of its mutational landscape and clinical implications.

The HBV surface antigen (HBsAg), encoded by the S gene, contains the “a” determinant (amino acids 124–147), a highly conserved region required for immune recognition, diagnosis, and vaccine development [20]. The Major Hydrophilic Region (MHR) of the HBV surface antigen, which contains the immunologically essential “a” determinant, has a significant impact on immune recognition and vaccine effectiveness. Mutations in the “a” determinant of the S gene (often called immune escape mutants—IEMs) can lead to a change in the structure of HBsAg, reducing antibody recognition [21]. During a chronic infection, this can result in immune and vaccine escape [22]. Furthermore, mutations in the S gene can affect the efficacy of HBsAg diagnostic tests, leading to false-negative results, particularly in cases of occult HBV infections (OBI) [23].

The basal core promoter (BCP)/precore (PC) region regulates the transcription and translation of the precore/core mRNA, which produces HBeAg [24]. Mutations in this region can impact HBeAg expression and influence disease progression [25]. Specific mutations in the BCP/PC region, such as the double mutation at positions 1762 and 1764 (A1762T/G1764A) in the BCP and at position 1896 (G1896A) in the PC region, can significantly reduce or abolish HBeAg expression [26]. These mutations are frequently observed in patients with chronic hepatitis and are linked to more severe liver disease. While HBeAg-positive infections often lead to immune tolerance, the emergence of HBeAg-negative mutants can trigger a more aggressive immune response, potentially accelerating liver damage, and increasing the risk of developing hepatocellular carcinoma [26]. The G1896A mutation halts the synthesis of Hepatitis B e-antigen (HBeAg) by replacing a tryptophan at amino acid position 28 with a premature stop signal and is among the most frequently seen PC mutations in chronic HBV cases [27,28].

Previous studies have documented the prevalence and transmission dynamics of HBV in sub-Saharan Africa, including Nigeria [6,12,13]. Despite the high prevalence of HBV in Nigeria, there is limited information on the molecular diversity of HBV in some regions. Furthermore, little is known about how genetic mutations in HBV vary geographically within Nigeria, as well as the implications for public health strategies. Understanding the prevalence and diversity of MHR, BCP, and PC region mutations in HBV is critical for developing assays and vaccination strategies. In this study, we used the S gene and the BCP/PC region to investigate the genetic diversity, phylogenetic relationships, and mutational profiles of HBV strains isolated from two Nigerian regions: Plateau State in the north-central zone and Osun State in the south-west zone. Furthermore, this study provides an update on the potential emergence and circulation of HBV vaccine escape mutants, immunoglobulin-resistant strains, and BCP/PC mutants among asymptomatic HBV carriers in two regions of Nigeria. These findings shed light on the molecular epidemiology of HBV in these two distinct geographical regions in Nigeria, adding to the growing body of evidence required to optimize vaccination strategies and diagnostic tools in high-endemic regions.

## 2. Materials and Methods

### 2.1. Study Location and Sample Collection

This study was conducted at the Plateau State Specialist Hospital, Plateau State (North-Central Nigeria), and Osun State University Teaching Hospital, Osogbo, Osun State (Southwestern Nigeria), to ensure demographic and geographic diversity. Prior to sample collection, ethical approval was obtained from the Plateau State Specialist Hospital, Osun State Ministry of Health Ethical Review Board, and UNIOSUN Teaching hospital (reference numbers: NHREC/09/23/2010b, OSHREC/PRS/569T/521 and UTH/REC/2023/10/809, respectively). Participants provided written informed consent, and confidentiality measures were implemented before sample collection. A structured questionnaire was administered to collect demographic, lifestyle, and clinical data. Blood samples (5 mL) were collected aseptically into EDTA vacutainer tubes, processed for plasma separation by centrifugation at 3000 rpm for 15 min, and stored at −20 °C for further analysis. A total of 450 asymptomatic patients were tested for HBV using a one-step HBsAg strip rapid detection kit (RDT) (Hangzhou Biotest Biotech Co., Ltd., Hangzhou, China, LOT HBSG22040020, Ref No: IHBSG-S31) of which 223 tested positive for HBV. Only patients who tested positive for HBV using RDT were included for further analysis. All participants included in this study had no previous history of treatment for HBV, as seen in their records at the respective hospitals.

### 2.2. Serological Assay

To further confirm the positivity of patients that were positive using the RDT kits and determine the immune kinetics of HBV among the cohort, the 223 HBsAg-RDT-positive samples were tested using HBsAg ELISA (Melsin Medical Co., Changchun, China), lot no. (X20230501). Subsequently, HBsAg-positive samples confirmed using ELISA were also tested for Hepatitis B virus e antigen (HBeAg) and Hepatitis B core-related Antigen (HBcrAg) using commercially available enzyme-linked immunosorbent assay (ELISA) kits (Melsin Medical Co., Changchun, China) lot nos. (KM20231101 and YBS16922).

### 2.3. HBV Genomic DNA Extraction and PCR Amplification of the Partial S and BCP/PC Gene

Total DNA was extracted from the plasma that was confirmed positive for HBsAg using ELISA using the Qiagen DNEasy kit (lot number 163049045; Qiagen, Hilden, Germany) with a 50 µL elution volume, according to the manufacturer’s instructions. Subsequently, molecular detection of HBV DNA was performed using a previously described nested PCR targeting a 400 bp fragment of the partial ‘S’ gene region [29,30], and 360 bp of the BCP/PC gene [31]. Briefly, the 400 bp fragment of the partial “S” gene region was amplified using the PuReTaqTM Ready-To-Go PCR Beads in strip tubes (Sigma-Aldricht, Darmstadt, Germany, CAT#GE27-9557-02) using primers HBV_S1F (5″-CTAGGACCCCTGCTCGTGTT-3’) and HBV_S1R (5′-CGAACCACTGAACAAATGGCACT-3′) for the first round and HBV_SNF (5′-GTTGACAAGAATCCTCACAATACC-3′) and HBV_SNR (5′-GAGGCCCACTCCCATA-3) for the second round of amplification. For amplification of the BCP/PC region, HBV DNA amplification was also performed using puReTaqTM Ready-To-Go PCR Beads in strips (Sigma-Aldrich, CAT#GE27-9557-02). The first and second rounds of PCR amplification were carried out using a 20 µL reaction with first-round PCR primers BCP_PC F1 (5-GCATGGAGACCACCGTGAAC-3) and BCP_PC R1 (5-GGAAAGAAGTCCGAGGGCAA-3), while the second-round PCR primers were BCP_PC F2 (5-CATAAGAGGACTCTTGGACT-3) and BCP_PC R2 (5-GGCAAAAAACAGAGTAACTC-3). The amplified PCR products were sized against a 100-base-pair (bp) molecular weight marker (New England Biolabs, Beverly, MA, USA) after they were resolved on 2% agarose gel stained with ethidium bromide and viewed using a UV transilluminator (Cleaver Scientific, Rugby, UK).

### 2.4. Sequencing of the S and BCP/PC Genes

All the amplicons of the secondary PCR reaction (second-round PCR amplification products) with the expected band size for S and BCP/PC genes, respectively, were selected for Sanger sequencing. The Sanger sequencing method involved an initial EXOSAP-IT clean-up where the PCR products were purified to remove excess primers and dNTPs from the secondary amplicons. The ExoSAP-IT^TM^ reagents were removed from −20 °C and kept cool throughout the procedure. For the EXOSAP-IT clean-up step, 5 µL of the amplicons was purified with 2 µL of ExoSAP-IT^TM^ (Affymetrix, Santa Clara, CA, USA, CAT# 78205) at 37 °C for 15 min and inactivated at 80 °C for 15 min. This reaction was then incubated at 80 °C for 15 min to inactivate the ExoS1q\AP-IT reagent. Subsequently, 4 µL of the big dye terminator ready reaction mix (BigDye™ Terminator v1.1 Cycle Sequencing Kit Life Technologies, Carlsbad, CA, USA, CAT 4337451), 7 µL of nuclease-free water, and 2 µL each of secondary PCR (HBV_SNF and HBV_SNR) and (BCP_PCF1 and BCP_PCF2) primers were added to the purified secondary amplicons. PCR amplification was performed in a thermocycler (Eppendorf VapoProtect Mastercycler pro, Hamburg, Germany) with a final volume of 20 µL. The cycling conditions for the amplification were 96 °C for 60 s, 30 cycles of 96 °C for 10 s, 50 °C for 5 s, and 70 °C for 4 min.

The sequencing products were further purified using the Big Dye Xterminator Purification kit (Life Technologies, Carlsbad, CA, USA, CAT# 4376487). The reaction contained 90 µL of SAM solution and Big Dye Xterminator bead solution per sample, which were vortexed for 20 min at room temperature. The sequencing was carried out on the Applied Biosystems 3500 XL Genetic Analyser (Applied Biosystems, Foster City, CA, USA) at the Institute of Genomic and Global Health (formerly ACEGID, Redeemer’s University, Ede, Osun State, Nigeria).

### 2.5. Bioinformatic Analysis

To determine which HBV group the Nigerian sequences belonged to, the S gene sequences were subjected to the online Genome Detective hepatitis B virus database (https://www.genomedetective.com/app/typingtool/hbv/ (accessed on 16 February 2025)). Subsequently, reference sequences of all human HBV genotypes (HBV genotypes A–H) were downloaded from the NCBI Virus database for phylogenetic analysis. The sequences were aligned using MAFFT version 7 and phylogenetic analysis was performed using IQ-TREE version 1.6.12 [32] with ModelFinder [33], and the tree was visualized using Interactive Tree of Life (iTOL) v6 [34]. We also examined all the S gene sequences generated in this study for mutations in the MHR and a-determinant domain. To determine substitutions in the BCP/PC gene sequences from this study, we aligned the sequences with reference wildtype (X75657.1) and mutant (KM606740) strains, respectively.

The HBV sequences generated in this study were submitted to GenBank under accession numbers PV295916-PV295997 (for S gene) and PV768822-PV768866 (for BCP/PC gene), respectively.

### 2.6. Statistical Analysis

The data obtained were analyzed using GraphPad Prism 10. Analyses were carried out using descriptive statistics with the chi-square and Fisher’s exact test, with *p*-values less than 0.05 used to determine statistical significance.

## 3. Results

### 3.1. Socio-Demographic Characteristics and Serological Profile of Respondents

In this study, of the 223 participants who tested positive for RDT screening, 44.8% (100) were from Jos in Plateau State (North-Central Nigeria), while 55.2% (123) were from Osogbo in Osun State (Southwestern Nigeria). The participants were between 16 and 65 years old with a mean age of 35.48 ± 11.10. Our analysis to confirm the positivity of the 223 patients who were HBsAg-positive by RDT using HBsAg, HBeAg, and HBcrAg ELISA screening showed that 79.8% (178) were confirmed positive for HBsAg using HBsAg ELISA (Table 1). Screening of the 178 samples that were confirmed positive for HBV for other serological markers including HBcrAg and HBeAg ELISA showed that 99% (176) and 38% (67) were positive for HBcrAg and HBeAg ELISA screening, respectively (Table 1). Furthermore, analysis of the HBV serological markers among the study cohort based on location, gender, age, and marital status showed that a high proportion of HBsAg-positive patients in Osun were equally positive for HBeAg, with females having a slightly higher proportion of HBV infection. We also observed that patients within the age groups 17–30 and 31–45 years, respectively, as well as those grouped in the single and married marital status groups, showed a high proportion of HBV serological markers (Figure 1A–D).

### 3.2. PCR Analysis, Sequencing, and HBV Genotype Distribution

All 178 ELISA-confirmed HBsAg-positive samples were subjected to viral DNA extraction, and surface and BCP/PC gene amplification by PCR. Only 95/178 (53.37%) samples were PCR-positive for the S gene PCR screen, while 91/178 (51.12%) were positive for BCP/PC gene amplification by PCR and were sequenced. Of the 95 S gene and 91 BCP/PC amplicons that were sequenced, only 76 and 60 sequences were exploitable for S and BCP/PC genes, respectively, and were subsequently used for HBV genotype and mutational analysis.

All the 76 contigs generated for the S gene were subjected to the HBV genotyping tool on the Genome Detective virus online tool and phylogenetic analysis. The HBV typing tool revealed that 75/76 (98.68%) belonged to genotype E and 1/76 (1.32%) belonged to genotype A (subgenotype A3). Our phylogenetic analysis results further confirmed the HBV typing tool results with sequences from this study clustering with the previously described HBV genotype E mainly from Africa, while the only genotype A detected in this study clustered with HBV subgenotype A3 (EU054331.1) that was detected in 2016 from a pregnant woman in Gabon (Figure 2).

### 3.3. Mutations Identified in MHR and the “a” Determinant Domain of S Gene Sequences

Alignment of the protein sequences of the partial surface gene of the 76 HBV sequences from this study alongside reference genotypes E and A revealed the presence of 14 amino acid substitutions in the MHR and the “a” determinant domain, of which three each were previously reported as vaccine escape mutants (VEMs) (L127P/R, S140T/L, and G145A), HBV immunoglobulin (HBIg) resistance (T131N, S143T, and W156R), and mutations reported in patients with reactivated infections (T115N, G159A/R, and F161Y), respectively (Figure 3).

### 3.4. Mutations Identified in the Basal Core/Precore (BCP/PC) Region

Alignment of the nucleotide sequences of BCP/PC PCR-positive samples led to the detection of various substitutions in the TA-rich genome regions, Kozak sequence, and classical BCP/PC variants (Figure 4 and Figure 5). Specifically, we identified a high proportion of C1741T in 34/42 (81%) along with the single A1762T or G1764A mutation in 14/42 (33%) and 18/42 (43%) as the dominant variants in the BCP region (Figure 6A–D). The predominant classical PC G1896A and G1899A variants were identified in 26/42 (62%) and 17/42 (40%) of participants in this study. Other PC variants, including C1950G and G1951T, were identified in 41/42 (98%) and 42/42 (100%) participants, respectively (Figure 6E–H).

The distribution of BCP and PC variants across HBeAg (+) and HBeAg (−) participants is shown in Figure 7A–E. A higher proportion of the BCP + PC variant was observed in HBeAg (−) individuals (88.9%) compared to HBeAg (+) individuals (33.3%), suggesting a potential association between this variant and HBeAg seroconversion or absence. Figure 7B–E further break down these findings by gender and age range within both HBeAg (+) and HBeAg (−) groups. Notably, the BCP + PC variant consistently dominated among HBeAg (−) individuals across both genders and most age ranges, while HBeAg (+) individuals showed a more mixed distribution, with varying contributions from the BCP and PC variants depending on gender and age, though BCP + PC was still present.

## 4. Discussion

In this study, we investigated the prevalence and distribution of HBV IEM, BCP, and PC variants in asymptomatic participants from North-Central and Southwestern Nigeria. One of our major findings was the presence of vaccine escape mutants and HBV immunoglobulin resistance variants among the asymptomatic participants who came for routine checkups in the various hospitals. Mutations in the “a” determinant domain that change the acidity or hydrophilicity of the amino acid loops frequently affect the structure of the amino acid loops as well as the antigenicity of HBsAg [35]. Although we identified two HBV genotypes (A and E), HBV genotype E was the most predominantly identified genotype, and it is still the dominant strain circulating in Nigeria [12,13,30,36,37].

This study detected various mutations in the major hydrophilic loop (target of neutralizing antibodies) including vaccine escape mutants (VEMs) (L127P/R, S140T/L, and G145A), HBV immunoglobulin resistance mutants (T131N, S143T, and W156R) (HBV immunoglobulin resistance mutants are variants that show reduced susceptibility to neutralization by hepatitis B immunoglobulin—these mutations alter the antigenic properties of HBsAg, reducing the binding affinity of antibodies and decreasing the effectiveness of HBIg [38,39,40] and mutations previously reported in patients with reactivated infections (T115N, G159A/R, and F161Y), respectively (Figure 3). Previous reports have documented the circulation of a repertoire of HBV IEMs among asymptomatic community dwellers and blood donors in Southwestern Nigeria and other West African countries [13,17,18,36,37]. To the best of our knowledge, this is the first report of HBIg variants and mutations linked to HBV-reactivated infections in asymptomatic individuals in Nigeria. Our report on the predominance of IEMs among HBV genotype E-infected cohorts from this study corroborates a recently reported genotype E IEM circulation in Southwestern Nigeria and other West African countries which constitute the HBV genotype E crescent [17,18,36,41]. However, this is contrary to previous studies globally that reported higher rates of IEMs in HBV genotypes A, B, and D [42,43,44].

We also identified new putative S gene mutations within and outside of the “a” determinant including I100L, L127P/R, P135H, S140T, S143T, W156R, G159R, F161Y, and L162Q for the first time in Nigeria. Although the most widely studied IEM is G145R and its variant G145A, other known immune-escape mutations within the “a” determinant including L127/PR, T131N, and S143T may also be responsible for changes in the loops’ structure and the antigenicity of HBsAg [20,35]. For instance, L127P/R and T131N mutations detected in this study have been reported in patients with reactivated infections [45,46], who are typically at risk because HBV persists in hepatocytes and other tissues as cccDNA [47]. These substitutions also correlate with reactivation mostly seen with HIV co-infection, where the immunosuppressive state induced by HIV exacerbates the effects of these HBV substitutions [48]. The weakened immune system is less capable of controlling HBV replication, increasing the likelihood of reactivation. HIV/HBV co-infection is common due to shared transmission routes and significantly alters HBV’s natural history. The immunosuppressive state leads to increased viral replication and the selection of specific mutations in the HBV genome [49]. However, cases of HBV reactivation due to escape mutants have been reported in a patient undergoing chemotherapy as well as cases of immunosuppression [45,46]. Furthermore, the presence of distinct antigenic differences between HBsAg of the vaccine strains and the prevalent HBsAg of the Genotype E circulating in West Africa has been reported [50]. Changes in residues within this region of the surface antigen may thus influence structural modifications that allow replication of the mutated HBV in vaccinated individuals. Thus, the spread of these HBV variants may have an impact on the effectiveness of diagnostic immunoassays and the success of HBsAg-based vaccinations [23]. Future studies in Nigeria aimed at analyzing the antigenicity and immunogenicity of these mutations in the major hydrophilic loop of these HBV variants are crucial for developing effective diagnostic tools and vaccination strategies.

We also observed a preponderance of BCP (A1762T or G1764A) and classical PC (G1896A and G1899A) variants across the cohort with the double BCP TA and PC variant predominantly seen among the cohort (Figure 6A–H). The BCP double mutations, particularly A1762T/G1764A, which were first discovered in HBV-positive Japanese patients [51,52], inhibit HBeAg production by reducing precore mRNA transcription while maintaining or increasing viral replication [53]. These mutations play a significant role in the progression of HBeAg-negative chronic HBV infection and have been linked to more serious liver conditions such as HCC [54,55]. The G1896A substitution, which adds a stop codon at position 28 in the Pre-C region, is another common Pre-C mutation linked to HBeAg negativity and has been frequently discovered to be associated with the codon 29 missense mutation G1899A. This study’s high prevalence of BCP/PC variants is consistent with previously reported prevalences among hepatocellular carcinoma and asymptomatic carriers in South Africa [56], and asymptomatic hospital attendees in Southwestern Nigeria [36]. Hence, this may explain the high prevalence of HBV among asymptomatic individuals in this study. Thus, increased active HBV surveillance is required to prevent the spread of these variants within and across the country.

We found a mixture of wildtype and double BCP TA and PC HBV variants across HBeAg (+) and HBeAg (−) cohorts, with the double BCP TA alone or in combination with PC most commonly associated with HBeAg (+) (Figure 7A–C). A similar mixed viral population of wildtype and PC and/or BCP variants in HBeAg (+) patients has previously been reported in a racially diverse population in North America, including among young children [57]. In addition, compared to HBeAg (+) males, HBeAg (+) females had slightly more dominant PC and BCP variants. Our findings contradict a previous study, which revealed that male HBV carriers are more likely to have BCP mutations, notably the A1762T/G1764A double mutation, and are more likely to develop more severe disease, including cirrhosis or HCC, due to the probable enhancement in HBV replication by testosterone [58,59,60]. We also observed that young people (aged 20–29) with both HBeAg (+) and HBeAg (−) showed a significant prevalence of PC or BCP variations. This finding is consistent with the report from Lau et al. [57] who discovered that PC and BCP variants were present and potentially dominant during the first two decades of life, with increasing prevalence over time [57]. These findings could suggest that people with BCP and PC variants are more likely to develop immune-active hepatitis B. However, Candotti et al. [16] revealed that the precore wildtype at position 1896 was a major risk factor for HBV genotype E maternofetal transmission, which continues to play a substantial role in the high frequency of chronic infection. The major limitation of this study was our inability to determine serum HBV-DNA levels to enable comparison between wildtype and PC and/or BCP variants in HBeAg (+) and HBeAg (−) patients. Thus, the further monitoring of our study cohorts is required to determine whether HBeAg (+) participants with dominant PC or BCP variants are more likely to develop active hepatitis B after HBeAg clearance. Additionally, the use of a nested PCR targeting a 400 bp fragment of the partial S gene and a 306 bp BCP/PC gene, which may have limited sensitivity, resulted in a low percentage of amplified sequences for both the S gene (76/178, 43%) and the BCP/PC region (60/178, 34%). This low amplification rate is similar to the 38% amplification rates previously reported in HBV-infected pregnant women from Ghana [16], although significantly lower than the 69% amplification rates reported in deferred donors from Guinea [18]. The small volume of the initial sample, the poor quality of the isolated DNAs, and denaturation during storage may also be contributing factors to the low amplification rate. The impact of these limitations includes potential underestimation of the prevalence of certain mutations and genotypes. The low amplification rates of the S region sequence (43%), which were used for the classification of HBV into genotypes, might also introduce a bias in the distribution of HBV genotypes A and E reported in this study. However, this is not too different from the previous studies reported in Nigeria, where Genotype E is the predominant genotype in the country [61]. A lower prevalence of genotype A has equally been reported in Nigeria [62] as opposed to other sub-Saharan African countries where HBV genotype A is reported more frequently (13% in Côte d’Ivoire and 33% in Cameroon) [61]. Future studies should consider optimizing the amplification system and determine the viral load prior to amplification to improve yield and accuracy.

## 5. Conclusions

In summary, we show that a repertoire of HBV IEMs circulates among asymptomatic outpatients in North-Central and Southwestern Nigeria who are predominantly infected with HBV genotype E. We also found a preponderance of BCP (A1762T or G1764A) and classical PC (G1896A and G1899A) variants in the cohort. These mutations have been linked to the progression of HBeAg-negative chronic HBV infection as well as more serious liver conditions. Thus, increased HBV surveillance is required to prevent the spread of these variants within and across the country, as well as to guide the implementation of effective vaccination strategies.

## Figures and Tables

**Figure 1 viruses-17-01101-f001:**
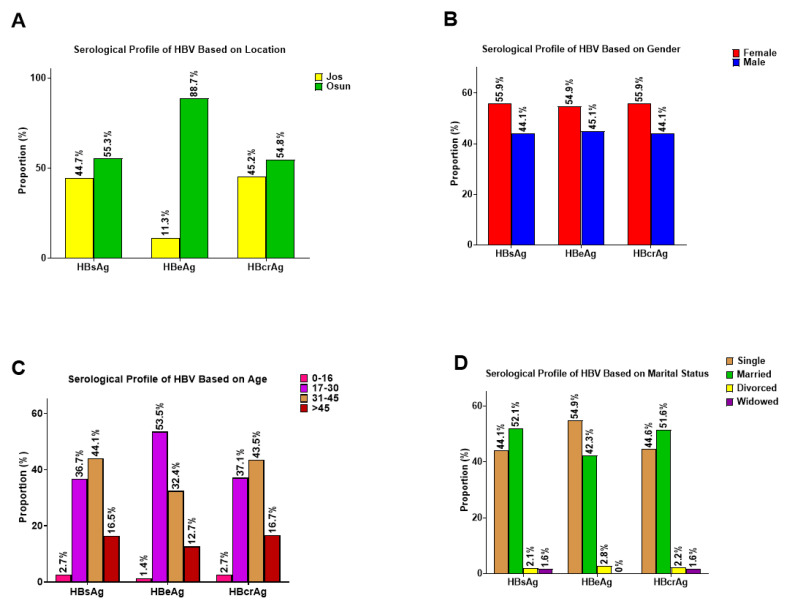
Serological profile of participants positive for HBsAg, HBeAg, and HBcrAg based on (**A**) Location; (**B**) Gender; (**C**) Age, and (**D**) Marital status.

**Figure 2 viruses-17-01101-f002:**
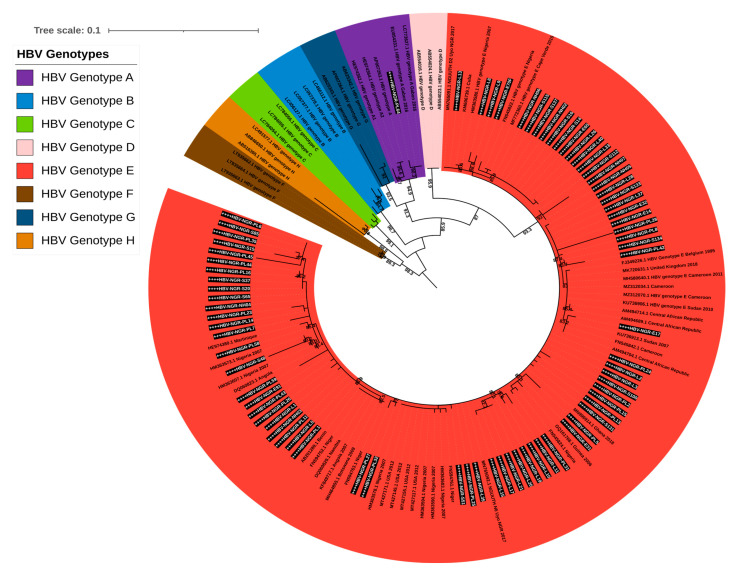
Maximum likelihood tree of HBV based on S gene sequences. Sequences reported in this study are asterisked and highlighted in white and black.

**Figure 3 viruses-17-01101-f003:**
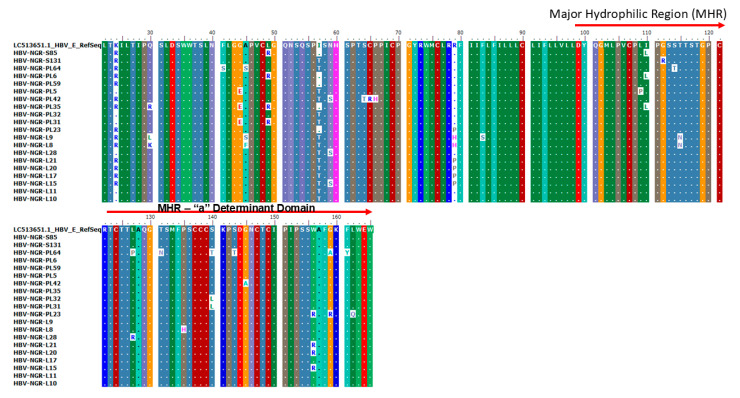
Amino acid alignment of the partial S gene sequences with the HBV genotype E reference sequence (accession number LC513651) indicating VEM, HBIg, and mutations reported in HBV reactivation at the MHR (amino acid positions 99–169) and the a-determinant domain found within the MHR (amino acid positions 124–147).

**Figure 4 viruses-17-01101-f004:**
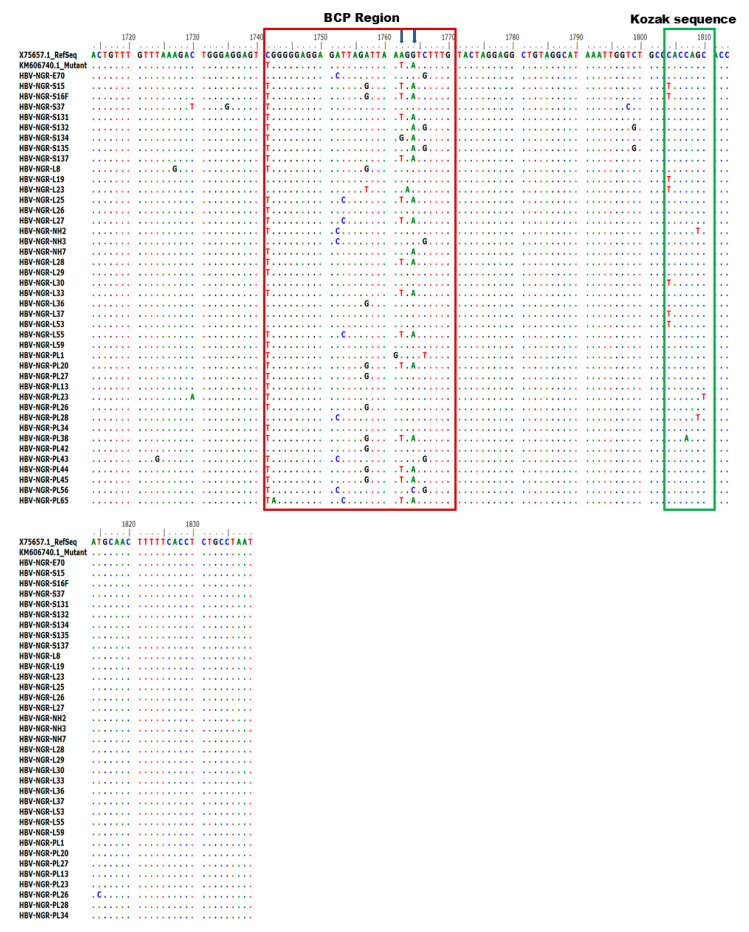
Alignment of partial BCP region gene sequences showing various mutations at the BCP region (highlighted in red) and Kozak sequence region (highlighted in green).

**Figure 5 viruses-17-01101-f005:**
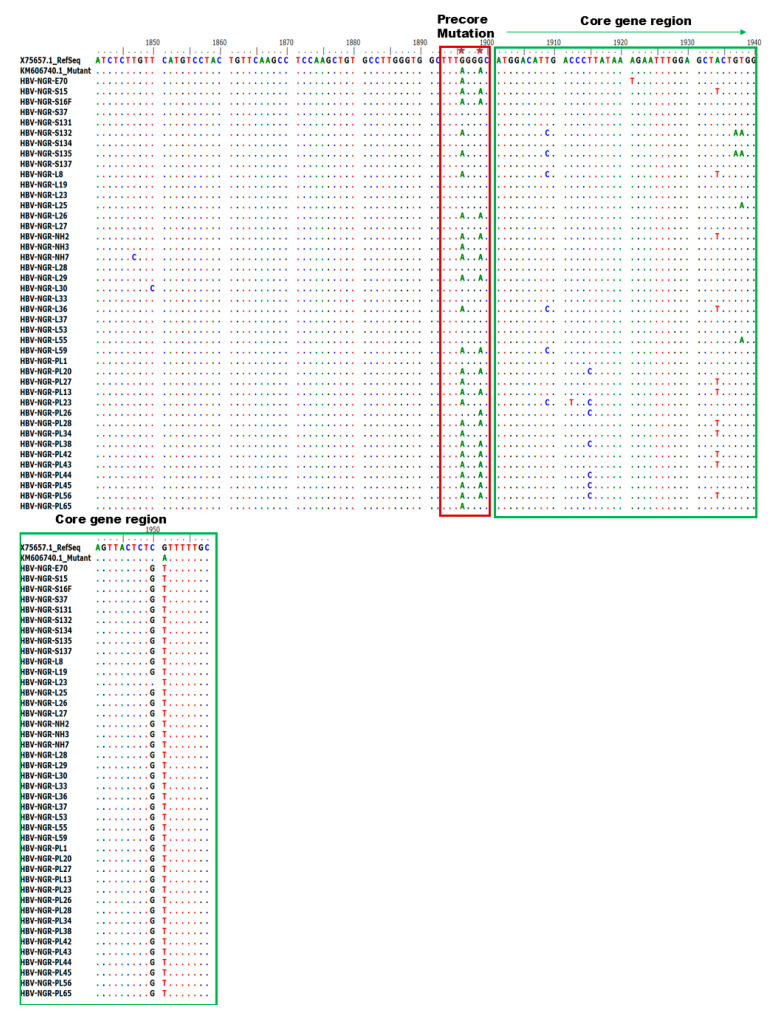
Alignment of PC region gene sequences showing various mutations at the core region.

**Figure 6 viruses-17-01101-f006:**
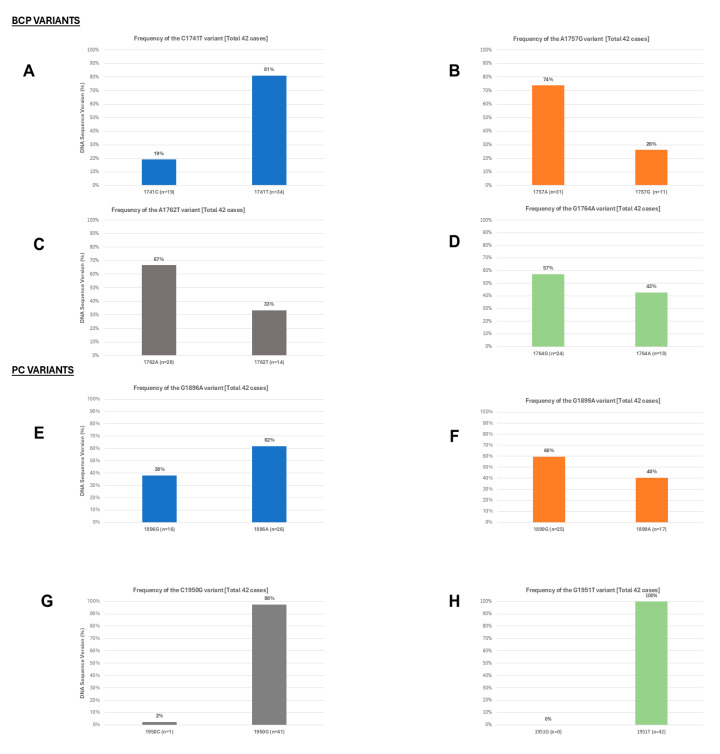
The frequency of dominant BCP and PC variants identified among the study participants, Based on frequency of variants in 42 cases (**A**) C1741T; (**B**) A1757G; (**C**) A1762T; (**D**) G1764A; (**E**) G1896A; (**F**) G1899A; (**G**) C1950G; and (**H**) G1951T.

**Figure 7 viruses-17-01101-f007:**
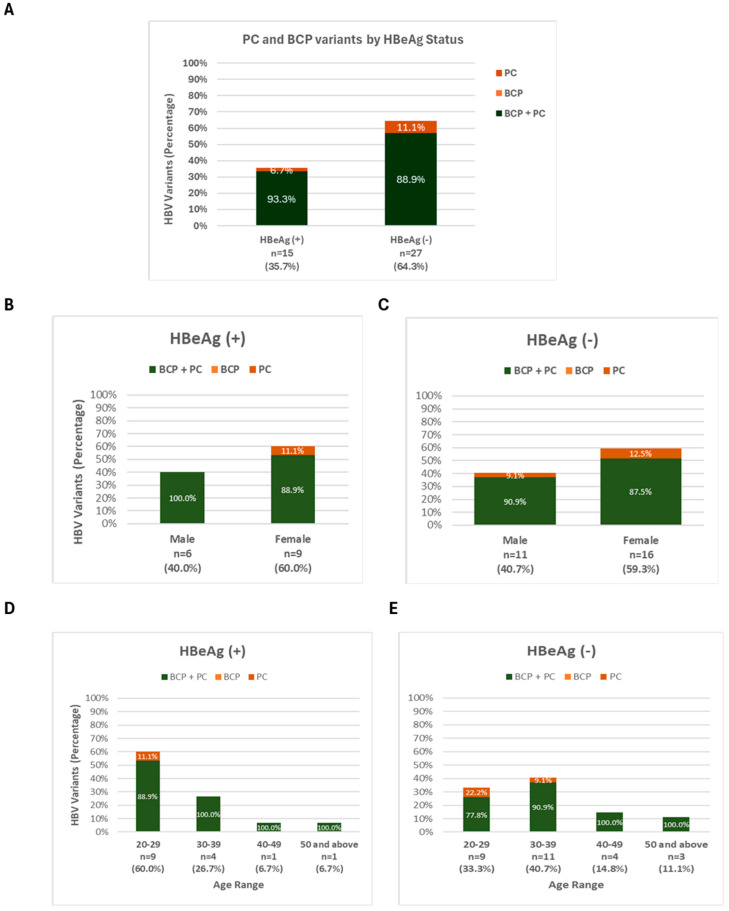
Prevalence of PC and BCP variants by (**A**) HBeAg status; (**B**,**C**) by Gender; and (**D**,**E**) by Age.

**Table 1 viruses-17-01101-t001:** Socio-demographic and serological profile of study population.

Characteristics		Frequency	(%)
Sex	Female	130	58.3
	Male	93	41.7
Location	Jos	100	44.8
	Osun	123	55.2
Age	0–16	5	2.2
	17–30	80	35.9
	31–45	100	44.8
	>45	38	17.0
Marital Status	Divorced	3	1.3
	Married	125	56.1
	Single	88	39.5
	Widowed	7	3.1
HBsAg	Negative	45	20.2
	Positive	178	79.8
HBeAg	Negative	111	62.4
	Positive	67	37.6
HBcrAg	Negative	2	1.1
	Positive	176	98.9

## Data Availability

The HBV sequences generated in this study have been submitted to GenBank under accession numbers PV295916-PV295997 (for the S gene) and PV768822-PV768866 (for the BCP/PC gene), respectively.

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
