# Peer review of "Evolutionary Insights of Hepatitis B Virus Genotypes and Profiles of Mutations in Surface and Basal Core Promoter/Pre-Core Genes Among HBsAg-Positive Patients in North-Central and Southwestern Nigeria"

_viruses, 2025, doi:10.3390/v17081101_

Round 1

Reviewer 1 Report

Comments and Suggestions for Authors

The manuscript by Abechi et al reports a study of HBV infections in two sites of Nigeria and results of sequencing of two regions of the genome, S and BCP/PC. Although partial, results are similar to previous studies carried out in Nigeria and other countries of West Africa where genotype E is dominant.

Introduction. Significant knowledge about HBV genotype E is available but was not quoted by the authors (Candotti et al J Vir Hep 2006; 89:715; Garmiri et al J Gen Virol 2009 90:2442; Candotti et al J Med Virol 2016; 88:2145) covering Guinea, Ghana and Burkina Faso. Data presented should be compared to these reports.

M&M. in 2.2, criteria for identifying confirm HBV infections should be provided.

Results. In Table 1, only the 178 cases of confirmed HBV infection should be presented; others should be considered false positive and excluded for presentation.

In figure 1, the vertical axis is not prevalence but presumably percentage of HBV reactive HBsAg RDT as in Table 1. Here again, only confirmed cases should be presented and analysed.

In section 3.2, the percentage of sequences obtained whether for S or BCP/PC is low and might be related to either or both low volume of initial sample (such volume should be given in M&M) or low viral load affecting amplification. The authors should provide a distribution of viral load in their population that may provide explanation for data obtained.

The poor yield of sequences (43% for S and 34% for BCP/PC) might introduce a bias in the distribution of genotypes since E genotype viral load tends to be significantly higher than genotype A. In other west African countries, genotype A appears more frequent than reported here (10% in Ghana, 25% in Burkina Faso and 5% in Guinea.

Discussion. The authors emphasise three S aa substitutions presumably related to HBV vaccination. It would be interesting to know whether individuals carrying such substitutions were anti-HBs positive or had been vaccinated.

Substitutions are also associated with ‘reactivated infections’. The authors should provide references supporting this categorisation and explain why such substitutions correlate with reactivation mostly seen with HIV co-infection.

Details should equally provide definition of what the authors mean by ‘HBV immunoglobulin resistance mutants’ with appropriate references and mechanism explanations.

Many other sequencing and identification of genotype E mutations in West Africa have been published and the authors should discuss how their data compare with those previous studies.

The main weakness of this study is the poor yield of both S and BCP/PC sequences probably related to small sample volume or low sensitivity of the nested amplification system used. The sensitivity of such assay should be indicated and the potential impact of such limitations be clearly stated and discussed.

Author Response

The authors appreciate the valuable suggestions and comments of the Editor and reviewers. These have certainly enriched the manuscript and have brought the best out of it. Below are our point-by-point responses to the reviewers’ comments.

Reviewer 1

Comments and Suggestions for Authors

The manuscript by Abechi et al reports a study of HBV infections in two sites of Nigeria and results of sequencing of two regions of the genome, S and BCP/PC. Although partial, results are similar to previous studies carried out in Nigeria and other countries of West Africa where genotype E is dominant.

Reviewers’ Comment 1

Significant knowledge about HBV genotype E is available but was not quoted by the authors (Candotti et al J Vir Hep 2006; 89:715; Garmiri et al J Gen Virol 2009 90:2442; Candotti et al J Med Virol 2016; 88:2145) covering Guinea, Ghana and Burkina Faso. Data presented should be compared to these reports.

Response:

We thank the reviewer for the thorough appraisal of our manuscript. As suggested, we have considered the content of the suggested articles and have incorporated the papers into the manuscript as additional information (lines 74 -78 in the introduction section).

MATERIALS AND METHOD

Reviewers’ Comment 2

 In 2.2, criteria for identifying confirmed HBV infections should be provided.

Response:

The criteria used for HBV confirmation was based on HBsAg ELISA positivity of which 178 out of the 223 HBsAg-RDT-positive samples were confirmed positive. We have also updated the MM section to capture this point in lines 141 - 147 which and now read “To further confirm the positivity of patients that were positive using the RDT kits and determine the immune kinetics of HBV among the cohort, the 223 HBsAg-RDT-positive samples were tested using HBsAg ELISA Melsin Medical Co., China Lot no (X20230501). Subsequently, HBsAg-positive samples were also tested for Hepatitis B virus e antigen (HBeAg) and Hepatitis B core-related Antigen (HBcrAg) using commercially available enzyme-linked immunosorbent assay (ELISA) kits (Melsin Medical Co., China Lot no (KM20231101 and YBS16922)”.

RESULTS

Reviewers Comment 3

 In Table 1, only the 178 cases of confirmed HBV infection should be presented; others should be considered false positive and excluded for presentation.

Response:

We have updated the manuscript as suggested as shown in the revised Table 1.

Reviewers Comment 4

In figure 1, the vertical axis is not prevalence but presumably percentage of HBV reactive HBsAg RDT as in Table 1. Here again, only confirmed cases should be presented and analysed.

Response:

We agree with the reviewer and we have revised the figure labels as well as the presentation of only confirmed positive HBV as suggested.

Reviewers Comment 5

In section 3.2, the percentage of sequences obtained whether for S or BCP/PC is low and might be related to either or both low volume of initial sample (such volume should be given in M&M) or low viral load affecting amplification. The authors should provide a distribution of viral load in their population that may provide explanation for data obtained.

Response:

We agree with the reviewer. However, due to limited funding for this project, viral load were not determined for the HBsAg-positive samples. We have included this as part of our study’s limitations (see lines 376 - 390 which reads “The major limitation of this study was our inability to determine serum HBV-DNA levels to enable comparison between wildtype and PC and/or BCP variants in HBeAg (+) and HBeAg (−) patients. Thus, further monitoring of our study cohorts is required to determine whether HBeAg (+) participants with dominant PC or BCP variants are more likely to develop active hepatitis B after HBeAg clearance. Additionally, we reported a low percentage of amplified sequences for both the S gene (76/178, 43%) and the BCP/PC region (60/178, 34%), which is significantly lower than the 69% amplification rates reported in deferred donors from Guinea [18], but similar to the 38% amplification rates previously reported in HBV-infected pregnant women from Ghana [16].

Reviewers Comment 6

The poor yield of sequences (43% for S and 34% for BCP/PC) might introduce a bias in the distribution of genotypes since E genotype viral load tends to be significantly higher than genotype A. In other west African countries, genotype A appears more frequently than reported here (10% in Ghana, 25% in Burkina Faso and 5% in Guinea.

Response:

We agree with the reviewer that the poor yield might introduce a bias in the distribution of HBV genotypes.. We have acknowledged this as one of the limitations of this study (lines 385 - 401 which reads “Additionally, the use of a nested PCR targeting a 400 bp fragment of the partial S gene and a 360 bp BCP/PC gene [29,31], which may have limited sensitivity [56], resulted in a low percentage of amplified sequences for both the S gene (76/178, 43%) and the BCP/PC region (60/178, 34%). This low amplification rate is significantly lower than the 69% amplification rates reported in deferred donors from Guinea [18], but similar to the 38%  amplification rates previously reported in HBV-infected pregnant women from Ghana [16]. The small volume of the initial sample, the poor quality of the isolated DNAs, and denaturation during storage could also be contributing factors to the low amplification rate. The impact of these limitations includes potential underestimation of the prevalence of certain mutations and genotypes. The poor amplification rates of the S region sequences (43%), which were used for the classification of HBV into genotypes, might also introduce a bias in the distribution of HBV genotypes A and E reported in this study. Nevertheless, genotype E is the most reported genotype in the country. In other sub-Saharan African countries, HBV genotype A has been reported more frequently (13% in Côte d’Ivoire and 33% in Cameroon) [57] than the proportion (1.3%) reported in this study. Future studies should consider optimizing the amplification system to improve yield and accuracy.

Discussion

Reviewers Comment 7

The authors emphasize three S aa substitutions presumably related to HBV vaccination. It would be interesting to know whether individuals carrying such substitutions were anti-HBs positive or had been vaccinated.

Response:

We appreciate the reviewer for this insight and acknowledge the importance of understanding the vaccination status and anti-HBs positivity of individuals carrying the three S amino acid (aa) substitutions related to HBV vaccination. However, we did not screen the study participants for anti-HBs. This underscores the need for ongoing surveillance and research to monitor the prevalence and impact of these mutations. We have already put in place a follow-up study that will include comprehensive analysis of HBV serological profile including vaccination and anti-HBs status data across the six geopolitical zones in Nigeria to better understand the relationship between these substitutions and HBV vaccination, which will help elucidate the mechanisms underlying vaccine-induced immunity and potential immune escape in the region.

Reviewers Comment 8

Substitutions are also associated with ‘reactivated infections’. The authors should provide references supporting this categorisation and explain why such substitutions correlate with reactivation mostly seen with HIV co-infection.

Response:

We agree with the reviewer that these substitutions correlate with reactivation mostly seen with HIV co-infection where the immunosuppressive state induced by HIV exacerbates the effects of these HBV substitutions (Karami et al., 2016). The weakened immune system is less capable of controlling HBV replication, increasing the likelihood of reactivation. HIV/HBV co-infection is common due to shared transmission routes and significantly alters HBV's natural history. The immunosuppressive state leads to increased viral replication and the selection of specific mutations in the HBV genome (Racheal et al., 2023). However, cases of HBV reactivation due to escape mutants have been reported in a patient undergoing chemotherapy  as well as cases of immunosuppression. (Wu et al., 2012; Salpini et al., 2015).

We have updated the manuscript to capture this point see line 336-344.

Reviewers Comment 9

Details should equally provide definition of what the authors mean by ‘HBV immunoglobulin resistance mutants’ with appropriate references and mechanism explanations.

Response:

We have updated the manuscript to capture this point see line 316-320.

Reviewers Comment 10

Many other sequencing and identification of genotype E mutations in West Africa have been published and the authors should discuss how their data compare with those previous studies.

Response:

We have updated the manuscript to capture the comparison of HBV genotype E mutations in our study with those previous studies. See line 320 - 325 which reads “Our report on the predominance of IEMs among HBV genotype E infected cohorts from this study corroborates a recently reported genotype E IEMs circulation in Southwestern Nigeria and other West African countries which constitute the HBV genotype E crescent [17, 18, 36, 38]. However, this is contrary to previous studies globally that reported higher rates of IEMs in HBV Genotype A, B and D [39, 40, 41]. ”

Reviewers Comment : 11

The main weakness of this study is the poor yield of both S and BCP/PC sequences probably related to small sample volume or low sensitivity of the nested amplification system used. The sensitivity of such assay should be indicated and the potential impact of such limitations be clearly stated and discussed.

Response:

We appreciate the reviewer for this suggestion and have updated the manuscript to capture the potential impact of such limitations as shown in lines 383 - 399 “Additionally, the use of a nested PCR targeting a 400 bp fragment of the partial S gene and a 306 bp BCP/PC gene, which may have limited sensitivity, resulted in a low percentage of amplified sequences for both the S gene (76/178, 43%) and the BCP/PC region (60/178, 34%). This low amplification rate is  similar to the 38%  amplification rates previously reported in HBV-infected pregnant women from Ghana (Candotti et al., 2007), although significantly lower than the 69% amplification rates reported in deferred donors from Guinea (Gamiri et al., 2009). The small volume of the initial sample, the poor quality of the isolated DNAs, and denaturation during storage may also be contributing factors to the low amplification rate. The impact of these limitations includes potential underestimation of the prevalence of certain mutations and genotypes. The low amplification rates of the S region sequence (43%), which were used for the classification of HBV into genotypes, might also introduce a bias in the distribution of HBV genotypes A and E reported in this study. However, this is not too different from the previous studies reported in Nigeria, where Genotype E is the predominant genotype in the country  (Forbi et al., 2013) . A lower prevalence of genotype A has equally been reported in Nigeria (Opaleye et al., 2022) as opposed to other sub-Saharan African countries where HBV genotype A are reported more frequently (13% in Côte d’Ivoire and 33% in Cameroon) (Forbi et al ., 2013). Future studies should consider optimizing the amplification system and determine the  viral load prior to amplification  to improve yield and accuracy”.

Reviewer 2 Report

Comments and Suggestions for Authors

The authors analyzed nucleotide sequences of 79 S genes and 60 BCP/PC genes isolated from HBsAg-positive asymptomatic carriers in Nigeria. The authors showed HBV genotype E is dominant in Nigeria and also found a preponderance of BCP (A1762T or G1764A) and 371 classical PC (G1896A and G1899A) variants in the cohort. The manuscript is well-written and easy to follow. Methodology looks correct and I think this paper can be published with minor modification. This type of investigation is important as genotype and mutation analysis provides valuable prognostic information that helps predict disease progression and treatment responses in HBV-infected patients. Furthermore, HBV escape mutants pose significant challenges in both diagnostics and clinical management.

Minor points:

  1. Line 166: Section 2.4, Please indicate catalogue number rather than lot number.
  2. Line 268: Please insert 42/42 before (100%).

Author Response

Reviewer 2 Comment 1:

Line 166: Section 2.4, Please indicate catalogue number rather than lot number.

Response: Catalogue number inserted

Reviewer 2 Comment 2:

Line 268: Please insert 42/42 before (100%).

Response:42/42 Inserted

Round 2

Reviewer 1 Report

Comments and Suggestions for Authors

The authors have successfully improved their report and taken into account each comment of this reviewer.